# Transducer Cascades for Biological Literature-Based Discovery

**Denis Maurel [1,\*], Sandy Chéry [1], Nicole Bidoit [2], Philippe Chatalic [2], Aziza Filali [2], Christine Froidevaux [2] and Anne Poupon [3,4]**

1 Laboratoire d'Informatique Fondamentale et Appliquée de Tours (LIFAT), Université de Tours, 37000 Tours, France; sandy.chery@univ-tours.fr
2 CNRS, Laboratoire Interdisciplinaire des Sciences du Numérique, Université Paris-Saclay, 91190 Gif-sur-Yvette, France; nicole.bidoit@lisn.fr (N.B.); philippe.chatalic@lisn.fr (P.C.); filali@lri.fr (A.F.); christine.froidevaux@lisn.fr (C.F.)
3 PRC, INRAE, CNRS, Université de Tours, 37380 Nouzilly, France; anne.poupon@inrae.fr
4 Inria Saclay-Île-de-France, Inria, Université Paris-Saclay, 91120 Palaiseau, France
\* Correspondence: denis.maurel@univ-tours.fr

**Abstract:** G protein-coupled receptors (GPCRs) control the response of cells to many signals, and as such, are involved in most cellular processes. As membrane receptors, they are accessible at the surface of the cell. GPCRs are also the largest family of membrane receptors, with more than 800 representatives in mammal genomes. For this reason, they are ideal targets for drugs. Although about one third of approved drugs target GPCRs, only about 16% of GPCRs are targeted by drugs. One of the difficulties comes from the lack of knowledge on the intra-cellular events triggered by these molecules. In the last two decades, scientists have started mapping the signaling networks triggered by GPCRs. However, it soon appeared that the system is very complex, which led to the publication of more than 320,000 scientific papers. Clearly, a human cannot take into account such massive sources of information. These papers represent a mine of information about both ontological knowledge and experimental results related to GPCRs, which have to be exploited in order to build signaling networks. The ABLISS project aims at the automatic building of GPCRs networks using automated deductive reasoning, allowing to integrate all available data. Therefore, we processed the automatic extraction of network information from the literature using Natural Language Processing (NLP). We mainly focused on the experimental results about GPCRs reported in the scientific papers, as so far there is no source gathering all these experimental results. We designed a relational database in order to make them available to the scientific community later. After introducing the more general objectives of the ABLISS project, we describe the formalism in detail. We then explain the NLP program using the finite state methods (Unitex graph cascades) we implemented and discuss the extracted facts obtained. Finally, we present the design of the relational database that stores the facts extracted from the selected papers.

**Keywords:** literature-based discovery; Finite State Methods; transducer cascades; Unitex; database design; automated deductive reasoning

## 1. Motivation

In a living cell, the so-called signaling network is the intricate biochemical network which triggers the adapted biological outcome in response to an external signal, such as the increase in concentration of a given hormone. A living cell is able to respond to thousands of different external signals in as many biological outcomes. These networks are involved in all physiological processes in the human body and consequently in all pathologies. In most cases, the first components in these cascades are membrane receptors. These molecules, which are part of the cell membrane, are specialized in detecting a signal: the concentration change of a given molecule, the presence of a nutrient and so on. These receptors are then capable of producing a specific signal inside the cell, which then triggers a whole cascade of biochemical interactions and reactions.

Membrane receptors, governing the cell's behavior and being accessible through general circulation, represent the best option for chemically modifying physiological processes, which is what drugs do. Indeed, about a third of available drugs target G-protein coupled receptors (GPCR) [1,2], which represent the largest class of membrane receptors in all mammals. Other drugs target other types of membrane receptors such as receptor protein kinases, ion channels or membrane transporters.

Understanding the biochemical wiring of the cell is a necessary step towards the rational design of new drugs. Knowing the network would allow us to decide what receptor can be used and what stimulation it should receive to obtain a desired effect. It would also help in understanding all the consequences of the stimulation of a given receptor: the main effect, which is the expected benefit, but also secondary effects.

In the last two decades, scientists have started mapping this wiring, especially the signaling networks triggered by GPCRs. However, it soon appeared that the system is very complex, which to the publication of more than 320,000 scientific papers (https://medium.com/swlh/gpcr-research-trends-fb827144da02, accessed on 1 April 2022). Clearly, a human cannot take into account such massive sources of information. This is precisely the objective of the ABLISS project: to be able to extract network information from the literature using Natural Language Processing (NLP), then build networks from these facts using automated deductive reasoning.

In a previous work, some of the authors manually extracted experimental facts from numerous papers and performed reasoning using some expert rules to infer new pieces of knowledge on the Follicle-stimulating hormone (FSH)-dependent and Epidermal Growth Factor(EGF)-dependent networks. Preliminary results were published in [3]. In order to generalize the approach, the need for an automatic extraction method of facts from scientific papers was crucial.

In our work, we chose to restrict information extraction from the "Results" section of large sets of papers. In biology literature, this section is dedicated to the factual description of experiments and their outcome, with minimal interpretation. Consequently, most of the facts extracted from this section should ensue from experiments described within the paper, as opposed to previous work, which is generally described in the introduction, or functional hypotheses, which will be made in the discussion section.

The work described in this paper presents the first steps of the project:

1. Formalization of biological facts.
2. NLP to extract facts from papers.
3. Building a database of predicates and facts.

## 2. Related Work

In biology, as in most areas of science, knowledge is shared in publications through raw text. As stated here above, even in seemingly specialized domains, such as GPCR signaling, the volume of these publications renders manual exploration impossible. Moreover, the vocabulary used in these publications is very heterogeneous. Indeed, Gregory Grefenstette has concluded from his studies that the probability that two experts use the same term for the same concept is only 20% [4]. To overcome this problem, ontologies were developed, starting in the 90s with EcoCyc [5], RiboWeb [6] and the well-known Gene Ontology [7]. Many different ontologies, some of them very specialized, have appeared [8], leading to the need for organizing and guiding the design of ontologies. In 2021, the OBO (Open Biological and Biomedical Ontologies) foundry [9] that establishes a set of principles for ontology development gathered more than 150 active ontologies.

While ontologies allow us to define a controlled vocabulary and annotating pieces of knowledge, the need to automatically extract these pieces of knowledge from plain text has proved to be crucial and has been met by Text Mining (TM). The first attempts relied on co-occurring terms to extract information concerning protein-protein interactions in abstracts [10]. Refinements of co-occurrence-based methods allow us to successfully extract this type of binary information (a simple relation between two named entities) [11]. To

extract more complex pieces of knowledge, improvements were gradually added. The first step towards NLP was the use of Link Grammar [12] and grammar-based parsers [13]. Different papers were then published illustrating the interest of NLP to analyze Biological texts [14,15], particularly for literature-based discovery [16,17]. More recently, efforts turned to deep-learning methods [18]. Although the results of simple tasks, such as information classification or even relation extraction, are very positive [19], the pathway extraction task, which is the object of the present work, is still problematic (F-score 0.37 [20]). The work that best meets our objectives is certainly INDRA [21], a software aiming to automatically build signal transduction pathways from text. INDRA uses various external NLP software systems offered as web services, with a focus on DRUM [22]. DRUM obtains a F-score of 0.57 when tested on selected passages of two scientific papers, which favorably compares with eleven other NLP software tested on the same selected texts. However, INDRA works on data extracted from public databases, and not on full-length scientific papers. Consequently, it can be used only to combine information already extracted from less structured sources and stored in a structured way. On the contrary, our objective is to extract information directly from scientific publications, which are highly unstructured and significantly larger than database extracts. To our knowledge, no equivalent work has been published.

## 3. Formalism

Using automated reasoning involves working with a precise formalism to describe components and interactions within the system. Different formalisms already existed at the beginning of this project, such as Petri-nets [23], logic-based models [24] and others, regrouped under the names *executable biology* or *algorithmic systems biology* [25–27]. However, none of these formalisms fulfilled the list of our requirements, which can be summarized as follows:

- The predicates and rules have to be both machine and human-readable. Moreover, they should be expressive enough for biologists to understand the meaning without much outside indication.
- The formalism should be chemically precise, meaning, for instance, that the different states of a given molecule have to be defined. In other words, the activation of a given molecule is not described as an *activation* but as the change of state from *inactive* to *active*.
- The direct and indirect actions need to be distinguished.
- The formalism has to allow the making of hypotheses.

In our formalism, we distinguish two types of predicates: *background* and *network* predicates. *background* predicates describe mainly the ontological and topological knowledge: the types of the components, the semantic relationships between the components and so on. *Network* predicates describe the effect(s) of a component on another one: how a given state of a first component can trigger a state transition of a second component.

### 3.1. Background Predicates
#### 3.1.1. Components

The first category of *background* predicates describes the different components of the system, shown in Figure 1. We chose not to use the SBO (Systems Biology Ontology) ontology [28], since we did not need all the complexity it offers. Instead, we focused on the components involved in signaling networks.

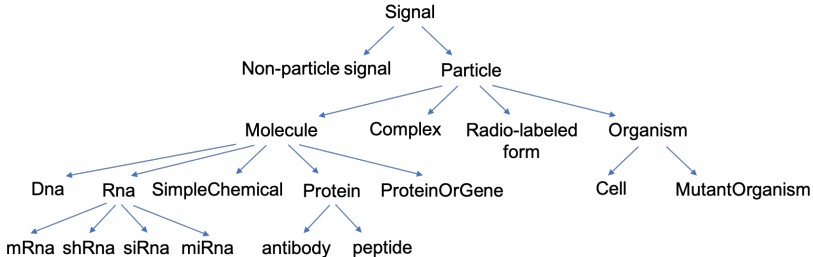

**Figure 1.** Ontology of the different types of components in a signaling network.

The highest level, *Signal*, encompasses all events that can have an effect on the system, either having physical existence (*Particle*), or not (*Non-particle signal*, such as temperature, concentrations of various elements and so on). A *particle* can be specified as a *molecule*, a *radio-labeled molecule*, a *complex* (which is a set of particles) or an *organism*. A *molecule* can further be refined in *RNA*, *DNA*, *simple molecule* or *protein*. The component *proteinOrGene* is used when the context does not allow us to distinguish between the two, since, in many cases, the protein and the gene coding for this protein have the same name. Finally, two types of proteins are individualized at level 5: *antibody* and *peptide*, as these have peculiar properties. For each named component, an ontological rule exists to go up one level: an antibody is a protein (*IF antibody(X) THEN protein(X)*), a protein is a molecule, a molecule is a particle and a particle is a signal and so on.

Molecules and complexes are the two types of component which can have different states. In our formalism, the state transition of a molecule reflects a chemical change of this molecule. For a protein, such a change can, for instance, be phosphorylation (the linking of a phosphate group on the protein) or a glycosylation (the linking of sugars on the protein). In most formalisms, these different states are not individualized, which is an important specificity of our methodology. For instance, as said before, in our formalism, the *activation* of a component is written as the transition from the *inactive* to the *active* state. This choice was made for two main reasons. First, a molecule has generally more than two states and the *active* status of a chemical form of the molecule depends on the function observed. An example is shown Figure 2: the component $X$ has three states, $X_1$, $X_2$, and $X_3$, each having a specific function: $X_2$ triggers the state transition of component $Y$ and $X_3$ triggers the state transition of component $Z$. In such case, talking about *activation* of $X$ is not sufficient, since this component has at least two active states. The second reason is that building the network of signaling system, which is what we intend to do in the present work, is not sufficient to fully understand its biological outcome and consequently for helping designing new drugs. Signaling networks are highly dynamical and their behavior can be simulated. The formalism we use leads to network models that can be readily used in dynamical simulation. It should be noted that depending on the type of molecule, the number of possible states is variable and often cannot be defined at the initial stage.

In fact, one of the objectives of studying a given system is precisely to:

- List the components of the system;
- Determine the different states of each component;
- Determine the actions each state triggers.

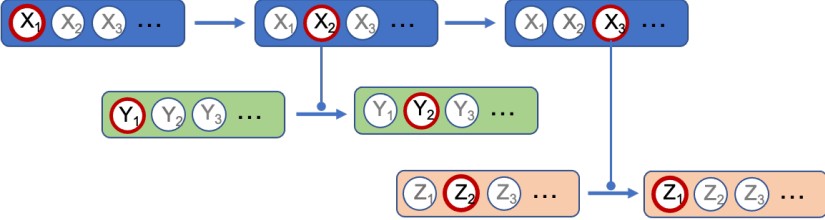

**Figure 2.** Example of a component having multiple active states.

As already mentioned, a protein can have different states and predicates have been introduced to link the different states, such as:

- Phosphorylated form, noted *phosphoForm(X,Y,P)*, which means *X is the protein Y, phophorylated at positions listed in P*. In the frequent case where the precise positions of phosphorylations are not known *P* takes the value *up* for *unknown position*.
- Acetylated form, similarly noted *acetylForm(X,Y,P)*.
- A more generic predicate *modifiedForm(X,Y)* is used to encompass other types of modifications, including those for which the precise chemistry is unknown, but change the properties of the protein, such as *active* and *inactive* forms. Ontological rules were written that allow us to identify *phosphoForm* and *acetylForm* as sub-categories of *modifiedForm*. Note also that *modifiedForm* is made symmetrical by the rule *IF modifiedForm(X,Y) THEN modifiedForm(Y,X)*.

The predicate *radioLabeledForm*, although very similar, reflects a different biological situation. Indeed, when we talk about a *phosphoForm* of a given component *X*, we designate a state of this component. However, radiolabeling, which is the process of grafting a radioactive atom on a molecule and is extensively used in experimental biology, modifies all the states of this molecule. Therefore, the result is a new component, as opposed to a new state. However, there is a strong relationship between the initial component and its radiolabeled form: they have the same states, each state having the same actions on the rest of the system and the transitions between the different states are due to the same causes. For these reasons, it was important to single-out this case.

The predicates *MutantGene*, *MutantProtein* and *MutantOrganism* are rather similar to *radioLabelledForm*, since they also generate new components: *MutantProtein(X,Y,L)*, means that *X is a mutated version of Y* (*L* being the list of mutations) and *Y* is a new protein, having the same states as *X* and the transitions between the different states of *Y* are triggered by the same events as for *X*. However, the properties of the different states of *Y* might be different form the properties of the corresponding states of *X*. This is precisely why mutations are often used in biological studies, since mutating a precise position of a protein can, for instance, abolish one of its properties without affecting the others and thus allow us to study the effects of this property on the rest of the system.

### 3.1.2. Other Types of *Background Predicates*

*Background* predicates are also used to describe the localization of the components, which is important for the functioning of the system. Finally, *background* predicates are used to define various other useful notions, such as:

- Abbreviations and alternative names;
- Methods;
- Organs;
- Lists: for instance, the predicate *particleList(L)* allows defining *L* as a list of objects of type *particle* and the predicate *particleListElement(L,X)* declares that *Y* is an element of that list;
- Predicate *transcribed(G,R)* links the gene *G* and the mRNA *R*, resulting from the transcription of *G*. Similarly, *translated(R,P)* links the mRNA *R* and the protein *P*, resulting from the translation of *R*.

### 3.2. Network Predicates

*Network* predicates indicate an action of a component on another component. An example is given Figure 3. In this example, we have two components, *Y* and *X*, which each have different states. We want to formalize the fact that state $Y_2$ of component *Y* is able to have an action on the state transition from $X_1$ to $X_2$ of component *X*. The predicate, *reactionModulation(Y2,X1,X2,E,D,S)* means that $Y_2$ modulates the state transition from $X_1$ to $X_2$, at some distance *D*, with the status *S* and the effect *E*. *E* gives the direction of this action and can take the values *increase, decrease* or *noeffect*, which means that *Y2* increases, respectively decreases, respectively has no effect, on the transition. *D* is the distance of this

action and can take the values *direct, indirect* or *unknown*. *S* is what we call the status of the predicate and can take the values *confirmed, hypothesis* or *bibliography*.

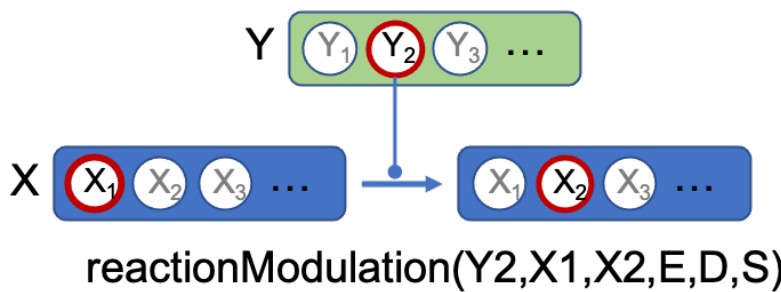

**Figure 3.** Predicate formalizing the fact that state $Y_2$ of component $Y$ has effect $E$ on the transition from state $X_1$ to state $X_2$ of component $X$, with distance $D$ and status $S$.

A status whose value is *confirmed* means that we have a proof, either experimental or through deduction using other confirmed predicates, that the fact is true. The status *hypothesis* means that we have no proof to back this predicate. However, as will be explained hereafter, in some cases the result of a single experiment can be explained by different and possibly mutually exclusive mechanisms and other experiments are necessary to select the relevant one. Thus, when considering solely the result of this single experiment, we will make the hypothesis that each of the possible mechanisms is true. Other rules, using further experiments, will then allow us to decide between the different mechanisms. Finally, the status *bibliography* is used when the authors of the analyzed sentence state that this fact comes from previous paper or work.

The *reactionModulation* predicate can be considered as a root predicate, describing a very simple mechanism. Other root predicates used in our system are:

- *localization(X,C,S)*, meaning that component $X$ is found in compartment $C$, with status $S$. The compartment used here corresponds to subcellular compartments: membrane, cytoplasm, nucleus and so on.
- *processModulation(X,P,E,D,S)* means that component $X$ has effect $E$ on the process $P$ at some distance $D$ and with the status $S$. Processes, which are defined in a dictionary, are biological processes, for example transcription or translation.
- *quantityModulation(X,Y,E,D,S)* means that component $X$ has effect $E$ on the quantity of component $Y$ at some distance $D$ and with the status $S$.

These root predicates can then be refined. Table 1 lists the refinements of the *reactionModulation* predicate. Similar refinements are made for the other types of root predicates. The precise list depends on the root predicate, since some refinements do not make sense, from the biological point of view, for some root predicates.

Finally, when extracting the predicates from the literature, two additional variables are included in all relevant predicates:

- *CL*, for *cell line*, which gives the cell line in which the experiment has been conducted;
- *MET*, for *method*, which gives the experimental protocol used.

These parameters will be present only in the initial facts extracted from the literature. They are not used anymore in the reasoning part since in many cases their values are too difficult to propagate through the rules.

**Table 1.** Refinements of the root predicate *reactionModulation*. For the sake of clarity we omit the comment on $D$ and $S$ in the second column, as they always have the same meaning.

| Predicate | Meaning |
|---|---|
| *reactionModulationCell-*($X_1$,$X_2$,E,$CL_1$,$CL_2$,D,S) | The transition from state $X_1$ to state $X_2$ is faster ($E$ = increase), respectively slower ($E$ = decrease), respectively equivalent ($E$ = noeffect), in cell line $CL_1$ than in cell line $CL_2$ |
| *reactionModulationMediation-*(M,$Y$,$X_1$,$X_2$,D,S) | The action of $Y$ on the transition from state $X_1$ to state $X_2$ is mediated by the component $M$, meaning that the action of component $Y$ on the transition from state $X_1$ to state $X_2$ of component $X$ is depending on the presence of component $M$. |
| *reactionModulationPerturbator-*($Y$,I,$X_1$,$X_2$,E,D,S) | The component $I$ increases ($E$ = increase), respectively decreases ($E$ = decrease, respectively has no effect ($E$ = noeffect, on the action of $Y$ on the transition from state $X_1$ to state $X_2$. |
| *reactionModulationSignalCell-*($Y$,$X_1$,$X_2$,E,$CL_1$,$CL_2$,D,S) | The effect of $Y$ on the transition from state $X_1$ to state $X_2$ is higher ($E$ = increase), respectively lower ($E$ = decrease), respectively the same ($E$ = noeffect) in cell line $CL_1$ than in cell line $CL_2$. |
| *reactionModulationSignalCompare-*(Y,Z,$X_1$,$X_2$,E,D,S) | The effect of $Y$ on the transition from state $X_1$ to state $X_2$ is higher ($E$ = increase), respectively lower ($E$ = decrease), respectively the same ($E$ = noeffect) than the effect of component $Z$ on the same state transition. |

One such case is when two predicates obtained in different cell lines are the premises of a rule, then the predicates in conclusion cannot be assigned to one of the two cell lines, nor to both. For example, if we know that $X$ triggers the state transition from $Y_1$ to $Y_2$ in cell line $CL1$, and that in cell line $CL2$, this action cannot be obtained in the absence of component $Z$, we can make the hypothesis that $Z$ mediates the action of $X$ on $Y$. However, we cannot decide whether this happens in cell line $CL1$, in cell line $CL2$ or in both. It would of course be possible to make the three different hypotheses by mentioning explicitly the cell lines concerned. However, it clearly appeared that this would drastically increase the complexity and thus we preferred to ignore the $CL$ parameter in the reasoning module. The same is true for experimental protocols. However, these variables were kept in initial facts, since they allow us to, for instance, filter initial facts relevant to a given cell line. Another usage is when we arrive at contradicting conclusions in the reasoning module, it is possible to trace back to initial facts leading to this contradiction and the protocols used. The user can then decide if one of the protocols is more reliable than the other.

*3.3. Rules*

The network inference module that uses the rules, will not be described in this paper. Still, it is important to rapidly present them, since some of the specific features of the predicates presented here above are linked to them. Here, again we have two types of rules: *Background* and *Network* rules. *Background* rules are further divided in two categories, namely *ontological* rules and *Typing* rules. *Ontological* rules link the different types of components in the hierarchy, for instance, the rule *IF particle(X) THEN signal(X)* means that if $X$ is a particle, then $X$ is also a signal. *Typing* rules specify the type of the arguments of the background predicates. For instance, the rules *IF radiolabeledForm(X,Y) THEN molecule(X)* and *IF radiolabeledForm(X,Y) THEN molecule(Y)* mean that if $X$ is a radiolabeled form of $Y$, then $X$ and $Y$ are molecules.

*Network* rules are designed to deduce new background facts and thus new pieces of knowledge on the studied network. An example of such a rule is given Figure 4. In

this example, the initial facts establish that state $Y_2$ of component $Y$, respectively $Z_1$ of component $Z$, have the effects $E_A$, respectively $E_B$ on the transition from state $X_1$ to state $X_2$ of component $X$. We can thus make the hypothesis that $Y_2$ has the effect $E_A * E_B$ on the transition from state $Z_2$ to $Z_1$ of component $Z$. Of course, this is not the only possible mechanism: for instance, $Z$ could have an action on $Y$ instead of the reverse and finally, the actions of $Y$ and $Z$ can be independent from each other. In addition, since biology is complicated, these hypotheses are not mutually exclusive.

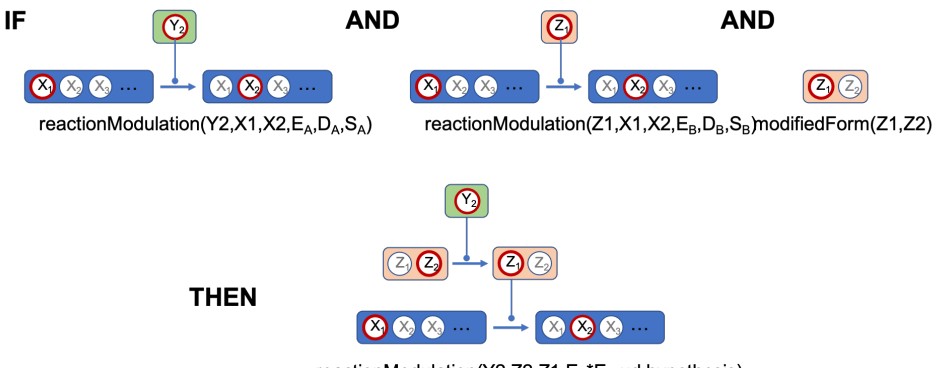

IF reactionModulation(Y2,X1,X2,E$_A$,D$_A$,S$_A$) AND reactionModulation(Z1,X1,X2,E$_B$,D$_B$,S$_B$) AND modifiedForm(Z1,Z2)
THEN reactionModulation(Y2,Z2,Z1,E$_A$*E$_B$,ud,hypothesis)

**Figure 4.** Schematic representation of a rule.

## 4. NLP to Extract Predicates

The objective of the work presented here is to extract a set of facts, using predicates, from natural language sentences found in scientific papers encompassing as much as possible the scientific information related by these sentences. A few examples are given in Table 2.

**Table 2.** Examples of sentences and corresponding facts. Abbreviations: HEK293, human embryonic kidney 293 cells; FSHR, follicle-stimulating hormone receptor; ERK, mitogen-activated protein kinase; $\beta$2AR, beta-2 adrenergic receptor.

| Sentence | Facts |
|---|---|
| In HEK 293 cells transiently expressing the rat FSH-R, FSH stimulated ERK phosphorylation in a dose-dependent manner (Figure 1 | cell(HEK293)<br>protein(FSHR)<br>transfectedCell(HEK293_FSHR, FSHR, HEK293)<br>protein(FSH)<br>protein(ERK)<br>phosphoForm(pERK, ERK)<br>reactionModulation(FSH, ERK, pERK, increase, unknownDistance, confirmed, HEK293_FSHR, unknownMethod) |
| Symetrically, overexpression of wild-type $\beta$-arrestin 1 or $\beta$-arrestin 2 enhanced the FSH-R internalization (Figure 5A). | protein($\beta$-arrestin 1)<br>protein($\beta$-arrestin 2)<br>protein(FSHR)<br>process(Internalization)<br>processModulationTargeted($\beta$-arrestin 1, Internalization, FSHR, increase, unknownDistance, confirmed, unknownCell, unknownMethod)<br>processModulationTargeted($\beta$-arrestin 2, Internalization, FSHR, increase, unknownDistance, confirmed, unknownCell, unknownMethod) |

**Table 2.** *Cont.*

| Sentence | Facts |
|---|---|
| These data indicate that ERK activation stimulated by the $\beta$2AR is mediated largely by $\beta$-arrestin isoforms. | protein(ERK)<br>modifiedForm(ERK,ERK_active)<br>modifiedForm(ERK,ERK_inactive)<br>protein($\beta$2AR)<br>reactionModulation($\beta$2AR, ERK_inactive, ERK_active, increase, unknownDistance, confirmed, unknownCell, unknownMethod)<br>protein($\beta$-arrestin)<br>reactionModulationMediation($\beta$-arrestin, $\beta$2AR, ERK_inactive, ERK_active, increase, unknownDistance, confirmed, unknownCell, unknownMethod) |

The NLP part of our project uses Finite State Methods presented in this paper. We use cascade of Finite State Transducers [29–31], where the transducers are Augmented Transition networks (ATN). We use the Unitex platform [32] to implement our cascades. In this framework, we can use variables to enrich the text by adding information and/or reorganize it by moving around some elements. Each cascade includes graphs, some of which are presented hereafter, for instance, at Figure 5. The Table 3 describes the notation used in Unitex graphs.

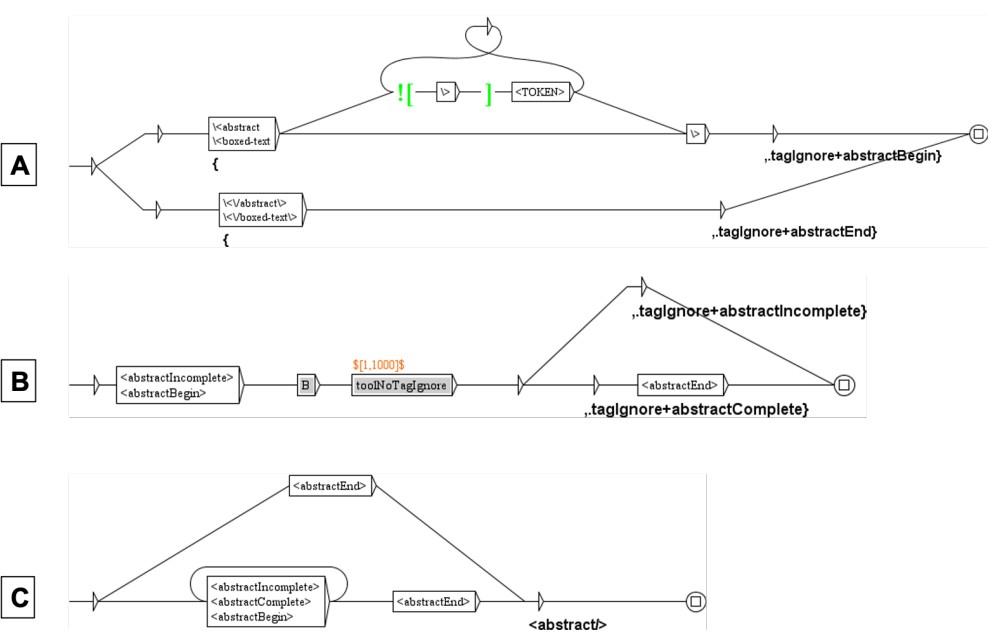

**Figure 5.** Graph (**A**) marks the beginning and the end of the abstract, recursive graph (**B**) transforms 1000 tokens of the text of the abstract in only one and graph (**C**) replaces the abstract by $<abstract/>$.

**Table 3.** Description of Unitex graphs.

| | |
|---|---|
|  | This graph should be read from left to right and recognizes the sequence *XX* |
|  | This graph recognizes all the sequences belonging to the category *CC* |
|  | This graph recognizes the sequence *XX* and merges it with *AA* (or replaces it by *AA*) |
|  | This grey box calls a subgraph named *SG* |
|  | If various paths are possible, the maximum matches is chosen: *XX YY* and not *XX* |
|  | This graph recognizes two sequences (*XX* and *YY*) and merges them in a new token *XX YY* whose category is *CC* with the optional feature *FF* |
|  | This graph recognizes the sequence *YY* except if the right context (beginning by *YY*) is *XX* |
|  | This loop recognizes all tokens until the right context *XX* |
|  | This loop can recognize one to ten times *XX* |
|  | For the same number of tokens matched, the path with the maximum last weight is chosen: *${2}$* |
|  | The parenthesis delimit a part of the recognized text that is stored in a variable *vv* (in superscript). This variable is used in the output (*$vv$*) |
|  | The box *$vv.SET$* tests if the variable *$vv$* is not empty; contrariwise the box *$vv.UNSET$* tests if the variable *$vv$* is empty |

The first step is to cut the "Results" section of each selected paper in individual sentences. We then extract from these sentences the background predicates (mostly components) and the experimental predicates, corresponding to actions of a component on another one.

### 4.1. Processing Chain

The processing chain to extract predicates is a PHP script that calls three Unitex cascade scripts. This chain is divided in three parts:

1. Preprocessing (Section 4.1.1)
   (a) A PHP script downloads papers selected beforehand based on a list of keywords.
   (b) An Unitex cascade script normalizes and reduces a paper to its "Results" section if it exists.
   (c) A PHP script eliminates papers in which the keywords used to for initial selection are not present in the results section.

2. Analysis (Section 4.1.2): A Unitex cascade script scans each paper and creates an XML-TEI file, structured paper by paper, paragraph by paragraph and sentence by sentence (Section 4.2.1).

3. Statistics (Section 4.2.2): A Unitex cascade script computes statistical results for the set of papers and for each paper and creates two output XML files.

### 4.1.1. Preprocessing

The starting point is the download of biological papers. For our proof-of-concept study, we are studying the GPCR-triggered pathway that depends on both *β*-arrestin and ERKs. The choice of this pathway relies on (1) the fact that the BIOS (Biology and Bioinformatics of Signaling Systems) group, to which the last author of this paper belongs, is internationally recognized in the domain of GPCR signaling, in particular for its studies on the role of *β*-arrestins. (2) Restricting to the *β*-arrestin/ERK pathway lead to an amenable size of the literature corpus. (3) Although ERK takes part in many signaling pathways (and not only GPCR-triggered), its exact role is not well known. We selected a corpus of publications using three keywords: *ERK, arrestin* and *phosphorylation*. We collected 5141 papers containing these three keywords in the full text.

The first Unitex cascade standardizes the text (spaces, dashes, apostrophes, suspension points...) and transcribes the HTML codes in Unicode characters (for instance, *&#x003B2;* to *β*). It then deletes the text before and after the "Results" section. This is not as simple as it can seem, since in some papers the abstract is a structured abstract, including a "Results" sub-section. Therefore, the first step is to delete the abstract to make sure that there is only one "Results" section. This deletion requires the three graphs shown Figure 5 (the deletion of the beginning and the end of the paper uses the same method), resulting in the replacement of the whole abstract by the tag $< abstract/ >$.

After the first Unitex cascade, we eliminate the papers without a "Results" section. We also eliminate all the papers which do not have the three keywords in the "Results" section. This reduces the number of papers to be analyzed to 548.

### 4.1.2. Analysis

This part is the core of the chain, in which we scan all papers and create an XML-TEI file with all the recognized predicates.

This cascade starts with preliminary treatments. First, parts of text that are not pertinent (editorial notes, footnotes, tables, figures and so on) are removed, using graphs similar to those presented Figure 5. Second, a graph splits paragraphs in sentences (Figure 6). We then replace contracted verbs by the developed form, replacing, for instance, *don't* by *do not*. We also develop numerated lists, replacing, for instance, *S1P1 to S1P3* by *S1P1, S1P2, S1P3* (Found in the sentence *In agreement, migration of MEFs, which express transcripts for S1P1 to S1P3, but not S1P4 and S1P5, toward S1P was increased when the S1P2 receptor was deleted*.) (S1PX: sphingosine 1-phosphate receptor X, S1P: sphingosine 1-phosphate).

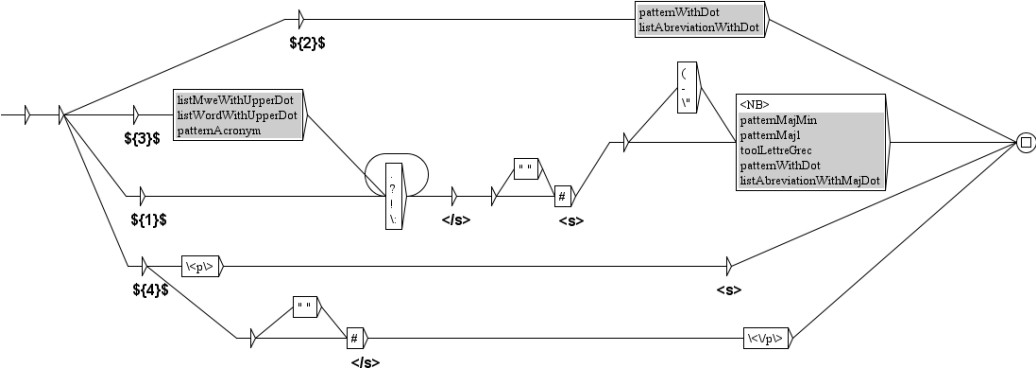

**Figure 6.** This graph splits the text in sentences and adds the TEI tags for sentences: $< s >$ and $< /s >$.

The cascade allowing the construction of the biological predicates can then be applied. One important feature of Unitex is that it allows the use of dictionaries. First, we use the Unitex English dictionary, which allows us, for instance, to tag verbs, adjectives and so on. Second, we use a dictionary of 277,387 proteins and 582,373 genes, built from public databases, to identify protein and gene names in the text. Finally, we use a specific custom dictionary containing 4041 entries which allows to tag entities specific to the study of biochemical networks, such as methods, actions and so on. This dictionary contains for instance:

- 1032 methods;
- 13 molecule families, which are expression used to designate a set of similar molecules, but are not present in the general dictionaries;
- 242 molecules that were added because they were not present in the general dictionaries;
- 111 localizations, corresponding to sub-cellular compartments;
- 1674 organisms;
- 93 biological process;
- 360 words or expressions describing an effect of a component on another one, which we gather under the term "regulations".

The graphs use these dictionaries from codes as *<protein>*, *<gene>* or *< methods>*. These dictionaries allow us to create ontological facts. For instance, in the detailed example (see Section 4.3), we recognize in a paper the sequence *ERK1/2* that appears in the protein dictionary, we tag it *protein(ERK1/2)* and we associate to this fact the type *background* and the subtype *ontological*.

We added a fifth dictionary used initially to eliminate the ambiguous entries between protein or gene on the one hand, and the English dictionary entries on the other hand. For instance, the entry *do* is a verb in the Unitex English dictionary but also a gene in the gene dictionary. In our work, we retain only the verb entry.

The next sixteen graphs tag expansions of proteins, as for example the graph of Figure 7 that distinguishes protein expansions and signal expansions. We then apply three graphs to expand also proteins, genes, signals, compounds, domains and complexes. The ambiguity between a protein and a gene is often resolved by the context of the sentence, and one of our graphs as been designed specifically for this task.

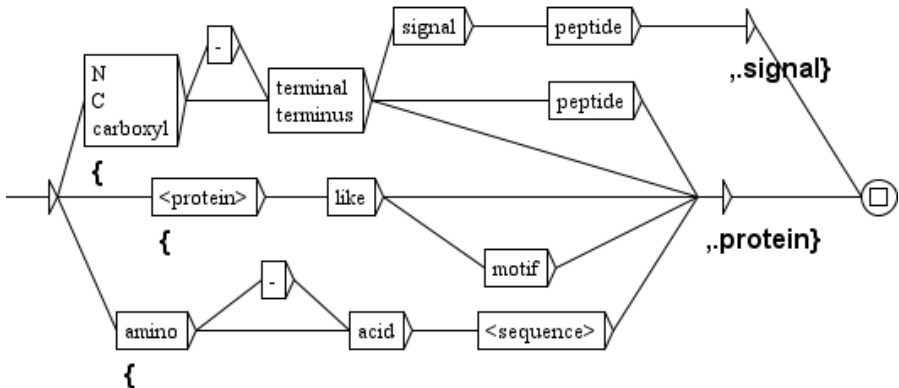

**Figure 7.** The graph to distinguish proteins from signals.

The goal of the graph presented in Figure 8 is very different. As we explained in the introduction of this section, we are above all interested in experimental results, to which we give the status *confirmed*, as opposed to other facts are annotated *bibliography* or *hypothesis*. The status *bibliography* can be deduced from the context of the sentence, either when there is an explicit reference to published work, or when author use expressions such as "in previous work". For instance: *As LiCl directly inhibits GSK3 kinase in the cytoplasm to stabilize of β-catenin* (Stambolic et al. 1996) (GSK3: glycogen synthase kinase-3) the inhibition fact comes from another paper. explains that the inhibition fact comes from another paper. In this case, the cascade will tag the predicates with the status *bibliography*. The status

*hypothesis* is sometimes also similarly deduced from the context, but can also be deduced from the verb, as we show below.

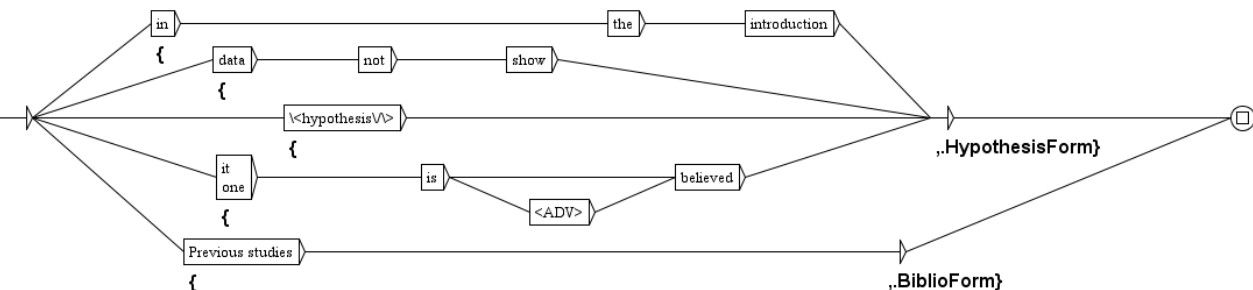

**Figure 8.** The graph to identify the status of an experimental fact.

The last step of this cascade, comprising three main graphs, builds the predicates in their final form. These are of course the most difficult graphs to build, since they extract the most complicated pieces of information. The initial versions of these graphs, which were used in the present work, were written by analyzing in detail eleven papers which do not belong to our GPCR/arrestin/ERK corpus.

The first task is to tag the background relation predicates. As of today, we have graphs for six such predicates: *acetylForm*, *deletedGene*, *isoform*, *knockOut*, *modifiedForm* and *phosphoForm*. These graphs tag the predicates and merge the lists of their arguments.

For instance, the graph of the Figure 9 tags the predicate *modifiedForm(X,Y)*, which means *X is a modified form of molecule Y*, whom arguments are typed *X:none* and *Y:molecule* (see Section 3). In the sentence of detailed example, *Pre-treatment with SR1 blocked ERK1/2 activation by both receptors.* (see Section 4.3), we recognize the protein ERK1/2 and the term "activation", leading to the construction of a *modifiedForm* predicate, tagged *modifiedForm(ERK1/2_active, ERK1/2_inactive)*.

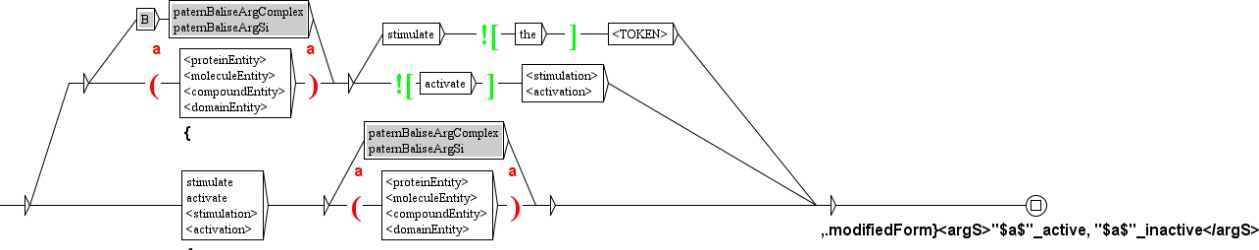

**Figure 9.** The graph to tag predicates of type *modifiedForm*.

We then tag regulation verbs (as defined in the specific dictionary): inhibition (block, inhibit, attenuate, decrease), activation (activation, stimulate, increase), expression (express, transcript, detectable, level), etc. At this stage, analysis of the verbs can be used to decide the status *hypothesis*. Indeed, the presence of verbs such as *may* or *could*, indicate that the authors are making an hypothesis. For instance, the sequence *could be directly inhibited* is tagged *hypothesis* by the graph of Figure 10.

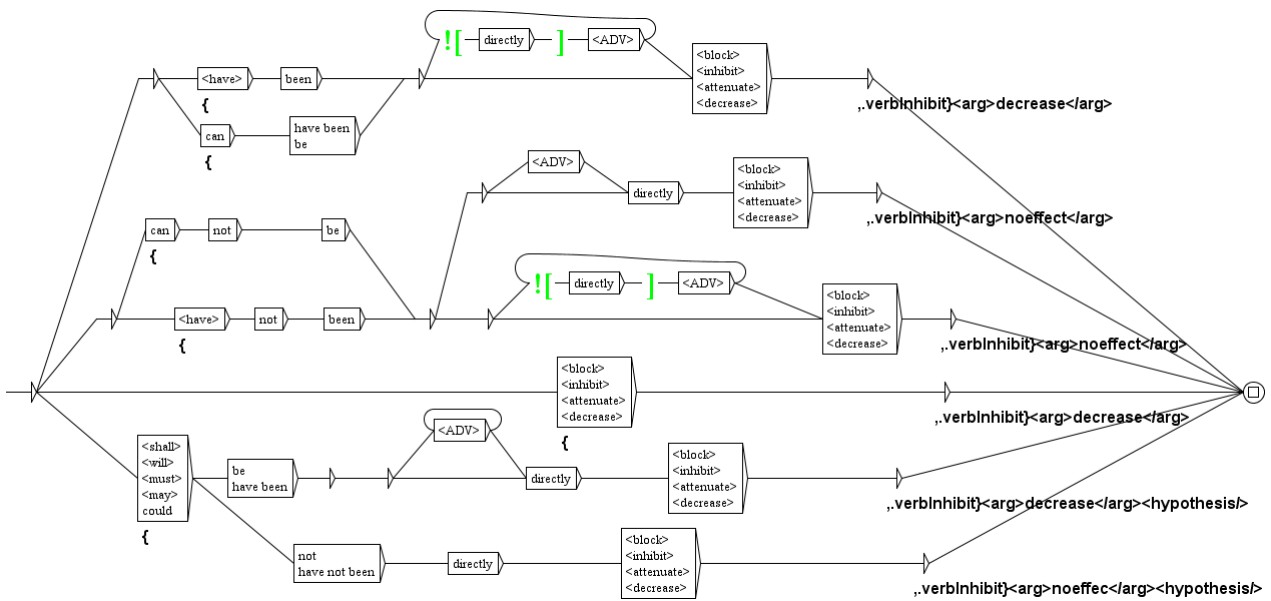

**Figure 10.** The graph to tag inhibition verbs.

The third step is to tag the network predicates. We have described eighteen of them: *association*, *associationCell*, *associationCompetition*, *associationModulation*, *associationModulation-Perturbator*, *associationModulationSignalCompare*, *cellBindingModulation*, *expressed*, *expressed-Cell*, *expressionCorrelation*, *localization*, *processTargetedCell*, *reactionModulationCell*, *reaction-ModulationMediation*, *reactionModulationMediation*, *reactionModulationPerturbator*, *transport-Modulation* and *transportModulationPerturbator*. As for the background predicates, these graphs tag the predicates and merge the lists of their arguments.

An example of such graph is given Figure 11. This graph tags the predicate *reactionModulationPerturbator(X,I,Y,Z,E,D,S,CL,MET)*, which means *Signal I modifies (increase, decrease, noeffect) the effect of X on the reaction Y -> Z, in cell line C*, whom arguments are typed *I:signal*, *Y:molecule*, *Z:molecule*, *E:increase,decrease,noeffect*, *D:diretc,indirect,unknown*, *S:confirmed,infirmed,hypothesis*, *CL:cell,organism,organ* and *MET:method* (see Section 3). In the sentence "*Pre-treatment with SR1 blocked ERK1/2 activation by both receptors.*" (see Section 4.3), we recognize the reactionModulationPerturbator predicate, tagged *reactionModulationPerturbator(SR1, unknownSignal, ERK1/2_active, ERK1/2_inactive, decrease, unknownDistance, confirmed, unknownCell, relation)* (SR1: Delta(14)-sterol reductase TM7SF2).

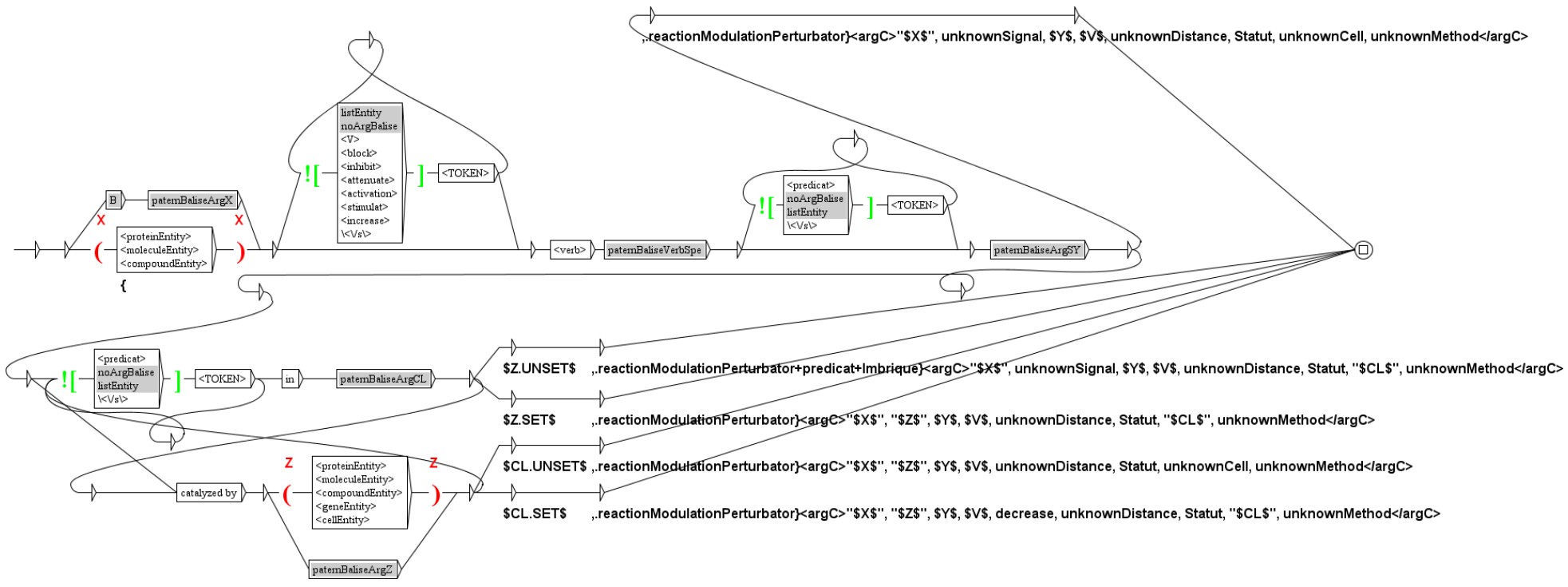

**Figure 11.** The graph to tag predicates of type *reactionModulationPerturbator*.

*4.2. Resulting Files*

4.2.1. XML-TEI File of Predicates

The resulting file of our extraction of predicates is a XML-TEI file that follows the structure below:

```
<teiCorpus>
    <teiHeader>ABLISS project presentation</teiHeader>
    <tei>
        <teiHeader>First text metadata</teiHeader>
        <text>First text</text>
    </tei>
    <tei>Following text</tei>
    ...
</ teiCorpus >
```

The text (restricted to the "Results" section ) is presented paragraph by paragraph and sentence by sentence. For each sentence, the extracted predicates are described:

```
<div type="paragraph">
    ...
    <div type="sentence">
        text of one sentence
        <desc type="background" subtype="ontological">predicate</desc>
        ...
        <desc type="background" subtype="relation">predicate</desc>
        ...
        <desc type="network" subtype="action">predicate</desc>
        ...
    </div>
    <div type="sentence">...</div>
    ...
</div>
```

4.2.2. XML Files of Statistics

Then we use the standOff cascade module of Unitex to extract statistics. We built two files: first, the global statistics of the ensemble of all texts and, second, the statistics text by text. These statistics files are XML-TEI files. The structure of the global statistics file is:

```
<tei>
    <teiHeader>ABLISS project presentation</teiHeader>
    <listAnnotation tag="desc" subtype="ontological">
        <ontological>
            <fact>gene("Amph")</fact>
            <frequency value="18"/>
        </ontological>
        ...
    </listAnnotation>
    <listAnnotation tag="desc" subtype="relation">...</listAnnotation>
    <listAnnotation tag="desc" subtype="action">...</listAnnotation>
</tei>
```

### 4.3. Detailed Example

To illustrate the complete process, we detail here an example: the sentence *Pre-treatment with SR1 blocked ERK1/2 activation by both receptors.* (see Figure 12) from [33].

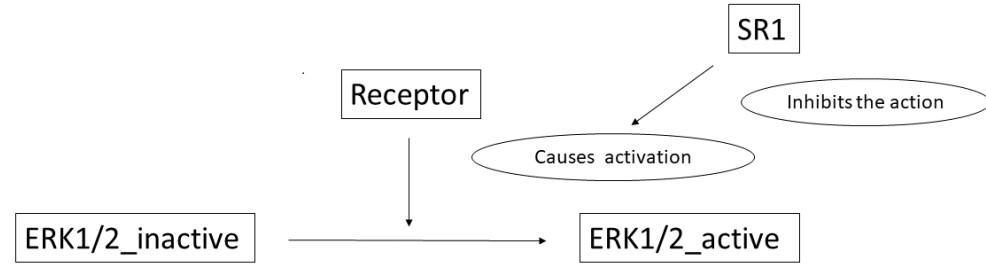

**Figure 12.** Pre-treatment with SR1 blocked ERK1/2 activation by both receptors.

The XML-TEI code resulting from the analysis of the complete paper is given below. The file starts with details on the paper itself, then gives the predicates found for each sentence. Only the part corresponding to the specific above sentence is detailed here.

```
<tei>
    <teiHeader>
        <titleStmt>
            <title>Rapid CB1 cannabinoid receptor desensitization defines the time course
of ERK1/2 MAP kinase signaling</title>
            <author>Tanya L. Daigle, Christopher S. Kearn, Ken Mackie</author>
        </titleStmt>
        <publicationStmt>
            <publisher>Neuropharmacology</publisher>
            <date>2005 May</date>
            <biblScope unit="vol">54</biblScope>
            <biblScope unit="issue">1</biblScope>
            <biblScope unit="pp">36</biblScope>
            <idno type="PMID">17681354</idno>
            <idno type="DOI">10.1016/j.neuropharm.2007.06.005</idno>
            <idno type="PMCID">PMC2277473</idno>
        </publicationStmt>
    </teiHeader>
    <text>
...
        <div type="paragraph">
            ...
                <div type="sentence">
                Pre-treatment with SR1 blocked ERK1/2 activation by both receptors.
                <desc type="background" subtype="ontological">
                    method(Pre-treatment)
                </desc>
                <type="background" subtype="ontological">
                    protein(SR1)
                </desc>
                <type="background" subtype="ontological">
                    protein(ERK1/2)
                </desc>
                <type="background" subtype="relation"
                    modifiedForm(ERK1/2_active, ERK1/2_inactive)
                </desc>
```

```
                <desc type="network" subtype="action">
                    reactionModulationPerturbator(SR1, unknownSignal,
ERK1/2_active, ERK1/2_inactive, decrease, unknownDistance, confirmed, unknownCell, un-
knownMethod)
                </desc>
            </div>
            . . .
        </div>
        . . .
    </text>
</tei>
```

*4.4. Evaluation*

To evaluate the performance of our *Processing chain*, we applied it to two papers in the list of 548 extracted using the keywords discussed previously. Note that these papers have not been used to build the graphs.

1. Flores 2014 [34];
2. Delgado 2016 [35].

We then evaluated the precision, recall and F-score by manually checking the predicates identified by the automated system. This evaluation was conducted by two biologists of our team. For each sentence, each predicate has been annotated as *true positive* or *false positive*. They also searched for missing predicates (*false negatives*). Then we computed Precision and Recall:

$$Precision = \frac{true positives}{(true positives + false positives)}$$

$$Recall = \frac{true positives}{(true positives + false negatives)}$$

There were few disagreements between the reviewers. These were discussed, and each time a consensus was easily reached. Indeed, most disagreements came from too quick reading by one or the other. For instance, in the sentence *Beta-arrestins were shown to coordinate multiple signaling networks downstream from many GPCRs*, the chain recognized the protein *Beta*, the protein *arrestins* and the complex *Beta-arrestins*; the first reviewer validated this, but the second said that *Beta-arrestins* as a whole was a protein, which is the correct interpretation.

During our analysis, we distinguished our ability to tag ontological predicates (Table 4) from the use of Unitex graphs to recognize relation and action predicates (Table 5). Indeed, the former is an easier task and leads to a high number of predicates, while the latter is more difficult and leads to a much lower number of predicates.

As for ontological predicates identification, our dictionaries fully cover our domain and we recognized all the ontological predicates: the chain has not produced false negatives (recall equals 100%). However, the precision is only 59.3%. We give a few examples:

- We found *protein(proteins)*, which is is a false positive. This error comes from a confusion between the tag *<protein>*, which designates a protein in our protein dictionary, and the lemma *protein* in the Unitex English dictionary. Thus, we will revise our graphs to avoid such a confusion.
- In the sentence *Negative regulation of sub-picomolar relaxin signalling requires PKA, PDE4 and β-arrestin 2* (PKA: cAMP-dependent protein kinase, PDE4: cAMP-specific 3′,5′-cyclic phosphodiesterase 4), we found only *protein(β-arrestin)* instead of *β-arrestin 2*. This comes from the fact that both terms are present in the dictionary of proteins, which is a good thing, since some authors do not specify which isoform has been used.

However, we have to take this into account in our graph, so that *β-arrestin 2* will have the precedence on *protein(β-arrestin)*.

- In the sentence *Thus, the Gα i3-Gβγ-PI3K-PKCζ pathway does not generate the cAMP detected by the pmEpac2 sensor* (*Gαi3*: guanine nucleotide-binding protein G(i) subunit alpha-3, *Gβ*: guanine nucleotide binding protein (G protein) subunit beta, *Gγ*: guanine nucleotide binding protein (G protein) subunit gamma, *PI3K*: Phosphatidylinositol 3-kinase, *PKCζ*: protein kinase C subunit *ζ*, cAMP: cyclic adenosine monophosphate), the list of proteins *complex(Gαi3 − Gβγ − PI3K − PKCζ)* was incorrectly identified as a complex. Indeed, the hyphen between two protein names can either mean a complexation between the two, or just a list, which can only be decided using the context. We will work on this problem, which is far from simple.

As illustrated by these three examples, beyond the simple evaluation of performance, this work gave us valuable insights for improving our graphs.

**Table 4.** Precision and recall for ontological predicates computed on two papers.

|  | True Positives | False Negatives | False Positives |
| --- | --- | --- | --- |
| Flores 2014 | 348 | 0 | 518 |
| Delgado 2016 | 879 | 0 | 322 |
| Total | 1227 | 0 | 840 |
|  | **Precision** | **Recall** | **F-Measure** |
| Flores 2014 | 40.2% | 100% | 57.3% |
| Delgado 2016 | 73.2% | 100% | 84.5% |
| Total | 59.4% | 100% | 74.5% |

**Table 5.** Precision and recall for relation and action predicates computed on two papers.

|  | True Positives | False Negatives | False Positives |
| --- | --- | --- | --- |
| Flores 2014 | 50 | 91 | 8 |
| Delgado 2016 | 33 | 144 | 13 |
| Total | 83 | 235 | 21 |
|  | **Precision** | **Recall** | **F-Measure** |
| Flores 2014 | 86.2% | 35.5% | 50.3% |
| Delgado 2016 | 71.7% | 18.6% | 29.6% |
| Total | 79.8% | 26.1% | 39.3% |

As for network predicates identification, we obtained a good precision (79.8%), but a poor recall score (26.1%). This poor recall score is partially explained by the the incompleteness of our list of predicates (see the list of implemented predicates in Section 4.1.2, to compare with the list of needed predicates in Section 3). For instance:

1. In the sentence *Together these results suggest that WIN induced prolonged activation of ERK1/2 in the mutant receptor that is solely mediated by β-arrestin 1*, a predicate *reaction-ModulationMediation* should in theory be found here, but the needed graph does not exist yet.

2. In the sentence *HEK293 cells expressing SEP-CB1Rs or S426A/S430A were cotransfected with β-arrestin 2 siRNA and exposed to WIN* (SEP: septin, CB1R: canabinoid receptor 1, WIN: WDR5-interaction inhibitor), the predicate *transfectedCell* should be found, but we have not yet created a graph for it.

As shown by these examples, implementing the missing graphs will lead to significant improvements of both precision and recall. Indeed, in the first paper, only 18 out of the 91 false negatives correspond to implemented recognition graphs. To improve precision, we should also lower the number of false positives. Here, again, analysis of the errors made in this first automated extraction gave us valuable leads for improvement. Firstly,

improvement of dictionaries and ontological predicates tagging will mechanically improve precision of network predicate tagging. Indeed, a predicate cannot be recognized if needed arguments have not be found in the sentence. For example, in the sentence *Lysates from untreated and agonist-treated (1 Î¼M WIN for 5 and 15 min) cells expressing the S426A/S430A receptor were applied to a nitrocellulose membrane spotted with antibodies for 43 kinases along with control antibodies.*, our graphs generated the fact *expressed("S-426A/S", ud, confirmed, "cells", um)*. We counted this as a false positive network predicate. However, the error comes from the fact that *S-426A/S* has been wrongly tagged as *molecule*, the correct tagging being *mutantProtein(S426A/S430A receptor)*. However, the graph corresponding to the *mutantProtein* does not exist yet.

## 5. From the NLP Extracted Facts to the Database

We have designed a SQL-relational database ABLISS-DB to store the background knowledge on signaling networks as well as network facts describing experimental results collected from the literature. The database schema is also suited for storing the expert inference rules needed for building the GPCR-triggered signaling networks from these facts, using deductive reasoning. Only the step concerning storage of facts is currently completed.

Our motivation for storing all the facts extracted by the transducer cascades method is manifold. First, network facts, as extracted from the "Results" section of scientific papers, are of major interest but unfortunately they are not stored in public databases. Secondly, building a database gives us the opportunity to specify the predicates of interest, providing both a formal description and a precise biological meaning. Finally, the database allows for some form of validation of the facts extracted by the transducer cascades method, when uploaded into the database.

### 5.1. Specification of Predicates

Facts are built using predicates with precise values for their arguments. Predicates correspond to those targeted by the biologists but, since scientific knowledge in this domain is constantly evolving, we kept the possibility of adding, changing or deleting predicates. Therefore, the schema allows for the description of predicates as first class citizen, with all their characteristics, according to the dictionaries designed by biologist experts. In some sense, the database maintains a dictionary of all the predicates of interest.

In our schema, predicates are described using three tables: `Predicate`, `PredArgument` and `PredArgType`. The `Predicate` table (Table 6) has 4 attributes that describe, for each predicate, its name, its arity, its type, and an extended description of its meaning (in natural language) with respect to its arguments. The `PredArgument` table (Table 7a) stores for each predicate, the rank and name of each of its arguments.

As an illustration, the specification of the predicates *protein(P)*, *modifiedForm(X,Y)* and *reactionModPert(X,I,Y,Z,E,D,S,CL,MET)* are represented in the database by the tuples listed in Tables 6 and 7a,b (where *reactionModPert* is here an abbreviation for *reactionModulation-Perturbator*).

**Table 6.** The `Predicate` table.

| PredName | Arity | Type | Description |
|---|---|---|---|
| modifiedForm | 2 | background | molecule X is a modified form of molecule Y |
| protein | 1 | background | P is a protein |
| reactionModPert | 9 | network | Signal I has effect E on the effect of X on the reaction Y -> Z, in cell line CL at the distance D, using method MET with the status S |
| … | … | … | … |

**Table 7.** Tables related to predicate arguments.

| (a) The `PredArgument` Table. | | |
| --- | --- | --- |
| **PredName** | **ArgRank** | **ArgName** |
| modifiedForm | 1 | X |
| modifiedForm | 2 | Y |
| protein | 1 | P |
| reactionModPert | 1 | X |
| reactionModPert | 2 | I |
| reactionModPert | 3 | Y |
| reactionModPert | 4 | Z |
| reactionModPert | 5 | E |
| reactionModPert | 6 | D |
| reactionModPert | 7 | S |
| reactionModPert | 8 | CL |
| reactionModPert | 9 | MET |
| … | … | … |
| … | … | … |

| (b) The `PredArgType` Table | | |
| --- | --- | --- |
| **PredName** | **ArgRank** | **ArgTypeId** |
| modifiedForm | 1 | AT1 |
| modifiedForm | 2 | AT1 |
| protein | 1 | AT2 |
| reactionModPert | 1 | AT3 |
| reactionModPert | 2 | AT3 |
| reactionModPert | 3 | AT1 |
| reactionModPert | 4 | AT1 |
| reactionModPert | 5 | AT4 |
| reactionModPert | 6 | AT5 |
| reactionModPert | 7 | AT6 |
| reactionModPert | 8 | AT7 |
| reactionModPert | 8 | AT8 |
| reactionModPert | 9 | AT9 |
| … | … | … |

The `PredArgType` table is used to describe the type of each argument. The motivation for representing separately argument types is due to the fact that, for some predicates, some arguments can have several types. For example, the 8th argument of reactionModPert, named CL, has two types namely, AT7 and AT8, which are *cell line* and *organ*. Indeed, biological experiments can be conducted either on cultured cells (*CL* will then designate a type of cell) or on biological tissues, extracted from an organ (*CL* will then designate this organ). Descriptions of the possible types for predicate arguments are stored in a separate table `ArgumentType` (Table 8).

**Table 8.** The `ArgumentType` table.

| ArgTypeId | Description | TypeName; Possible Values |
| --- | --- | --- |
| AT1 | a set of atoms (at least two) | molecule |
| AT2 | a long chain of amino acid residues | protein |
| AT3 | a signal | signal |
| AT4 | result of the action of some process | effect; choice: increase, decrease, noeffect |
| AT5 | specifies how far the action is performed | distance; choice: direct, indirect, unknownDistance |
| AT6 | the confidence into the fact | status; choice: confirmed, biblio, hypothesis |
| AT7 | cell lines as listed in the dictionary | cell line |
| AT8 | list of organs from the dictionary | organ |
| AT9 | list of experimental methods from the dictionary | method |
| … | … | … |

This approach enables to add at any time a new predicate (a new line in the Predicate table), by specifying its arity and giving its description, using the already described argument types, or by adding new kinds of argument types and describing them. Similarly, existing predicates can be modified.

*5.2. Storing Facts*

Facts obtained from the NLP extraction are automatically stored in two tables named respectively `BackgroundKnowledge` and `NetworkFacts`. `NetworkFacts` can contain the description of facts coming from the NLP extraction as well as facts deduced by rules, using the reasoning module (not yet completed). In that case, we store the rules used to obtain these facts. The *Extracted* attribute indicates whether the network fact has been extracted (value yes) or deduced by rules (value no). In addition, for each extracted fact, we provide metadata by mentioning the source it comes from, that is, the bibliographic article from which it has been automatically extracted. A table `Source` describes the title of the paper, the authors and the name of the journal.

Adding a fact into the database requires multiple verification steps. First, we check whether the predicate with which the fact is built already exists in the `Predicate` table. If this is the case, we check that the number of arguments of the fact corresponds to the predicate arity. Further verification on the adequacy of argument types can then be performed. When everything succeeds a new fact is added in the tables. Facts that do not comply with existing predicates specifications are gathered separately and a warning is issued for user feedback.

If the predicate underlying the new fact does not exist in the table, it first has to be added as a new entry into the `Predicate` table with its name, its arity, its family and its description. Then we can proceed to the addition of this new fact. In the following we illustrate the tables with the facts of the detailed example of Section 4.3. We have two background knowledge facts: *protein(SR1)* and *modifiedForm(ERK1/2_active, ERK1/2_inactive)*, and one network fact: *reactionModulationPerturbator(SR1, unknownSignal, ERK1/2_active, ERK1/2_inactive, decrease, unknownDistance, confirmed, unknownCell, unknownMethod)*, for which we have described the underlying predicates (Table 9).

**Table 9.** Tables relative to facts.

| **(a)** `BackgroundKnowledge` | | **(b)** `NetworkFacts` | | | **(c)** `SourcesOfFacts` | |
| --- | --- | --- | --- | --- | --- | --- |
| **idFact** | **PredName** | **idFact** | **PredName** | **Extracted** | **idFact** | **idSource** |
| bkF1 | modifiedForm | nf1 | reactModPert | yes | bkF1 | 17681354 |
| bkF2 | protein | … | … | … | bfF2 | 17681354 |
| … | … | … | … | … | nf1 | 17681354 |
| … | … | … | … | … | … | … |

The *idSource* can be a PMID (Pubmed ID) or a reference in a dictionary, depending on the provenance of the fact. Another table describes the bibliographic source (not shown here). Note that several articles can provide us with the same fact, thus reinforcing our confidence in the fact. Finally the `ArgValuesFacts` table gives the values of each argument of the predicate for each fact. We show below the lines of the `ArgValuesFacts` table corresponding to the three facts we have selected (Table 10).

**Table 10.** The `ArgValuesFacts` table.

| idFact | PredName | ArgRank | Value |
|--------|----------|---------|-------|
| bkF1 | modifiedForm | 1 | ERK1/2_active |
| bkF1 | modifiedForm | 2 | ERK1/2_inactive |
| bkF2 | protein | 1 | SR1 |
| nf1 | reactModPert | 1 | SR1 |
| nf1 | reactModPert | 2 | unknownSignal |
| nf1 | reactModPert | 3 | ERK1/2_active |
| nf1 | reactModPert | 4 | ERK1/2_inactive |
| nf1 | reactModPert | 5 | decrease |
| nf1 | reactModPert | 6 | unknownDistance |
| nf1 | reactModPert | 7 | confirmed |
| nf1 | reactModPert | 8 | unknownCell |
| nf1 | reactModPert | 9 | unknownMethod |
| … | … | … | |

A graphical representation of the relational tables introduced in this section is given in Figure 13. For each table, underlined attributes compose the key when it is a strict subset of attributes. Blue arrows are used to show foreign keys where arrow's head points to the referenced attribute(s). Finally, arrows crossing a triangle are meant to capture ISA relationships (inheritance).

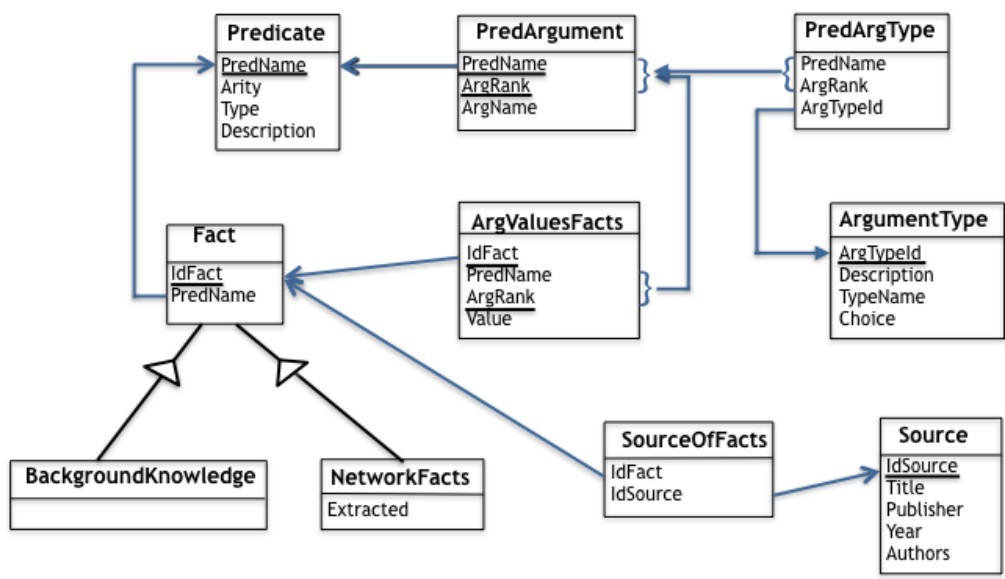

**Figure 13.** Graphical representation of relational tables.

## 6. Conclusions and Future Work

Our objective was to extract experimental results from scientific papers automatically. We concentrated on the "Results" section of papers, since it contains the experimental results obtained by the authors, as opposed to previous work (state of the art), mostly found in the introduction section. We also dismissed the discussion sections, as this section is dedicated to interpretation of the experimental results in light of previous work, and attempts to put them back in the more general context of the studied process. By doing so, we ensure that there is as little interpretation of the results as possible, and thus less bias. To this aim, we defined a new formalism suiting our needs, which we have introduced in this paper. To extract facts (instantiated predicates) from scientific texts, we used Finite State Methods, through Unitex graph cascades. The graphs were written based on eleven papers, and on a specialized dictionary that we have created. Finally, we have designed a

database to store (1) the predicates and (2) the extracted facts, from the output of the NLP program.

Our formalism includes 130 relation and action predicates. However, at present only 24 are described in our Unitex charts, and we plan to continue in this direction. Even if many graphs are still missing, we obtained a F-measure of 39.3%, as evaluated independently on two papers, which compares favorably with existing methods. This evaluation also led us to propose several tracks for improving the results, both on recall and on precision.

Beyond the trails discussed in the evaluation section, another way of improving our results is to consider the texts not only sentence by sentence, but as a whole. Indeed, as was shown in the different examples, when studying a single sentence, many attributes remain unidentified. For example, let us consider the already cited sentence *Together these results suggest that WIN induced prolonged activation of ERK1/2 in the mutant receptor that is solely mediated by β-arrestin 1*. Although we will be able to obtain the fact *reactionModulation(WIN, ERK1/2_inactive, ERK1/2_active, increase, unknownDistance, confirmed, unknownCell, unknownMethod)*, we will not be able to instantiate the attributes of the cellular context and the method, since the information is not in the sentence itself. To overcome this issue, we intend to implement a tool allowing the study of coordinations and anaphora. This will allow us to extract information missing in a sentence from previous sentences, for example, in the same paragraph, as sentences concluding on a experimental result are often found at the end of the paragraph precisely describing the experimental context and protocols.

In the future, the ABLISS database will also store the logical rules, needed for automated network inferences, as well as the deduced facts. A user-friendly interface will be developed to allow the continuous addition of new experimental facts and expert rules, as well as the querying of the database by biologists.

**Author Contributions:** Conceptualization, D.M., N.B., P.C., C.F. and A.P.; methodology, D.M., N.B., P.C., C.F. and A.P.; software, D.M. and S.C.; validation, D.M., S.C., N.B., P.C., C.F. and A.P.; formal analysis, D.M., S.C., N.B., P.C., A.F., C.F. and A.P.; writing, D.M., S.C., N.B., P.C., C.F. and A.P.; supervision, D.M., C.F. and A.P.; project administration, A.P. All authors have read and agreed to the published version of the manuscript.

**Funding:** Supported by ANR-18-CE45-0003-03 project (Abliss).

**Institutional Review Board Statement:** Not applicable.

**Informed Consent Statement:** Not applicable.

**Data Availability Statement:** Not applicable.

**Conflicts of Interest:** The authors declare no conflict of interest.

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
