# Peer review of "Transducer Cascades for Biological Literature-Based Discovery"

_information, doi:10.3390/info13050262_

Round 1

Reviewer 1 Report

The methodology is not clearly explained and described in the paper. Also, the related work must be definitely improved in the paper. As a result, I didn't see any contribution and the paper should not be accepted with the current version.

Reviewer 2 Report

I have one major but easy to resolve suggestion. I also have a few minor suggestions and many suggestions related to language choice, which should help improve the clarity of the paper. Feel free to use different wording to my suggestions. I have provided line numbers for each suggestion to speed up the revision process.

Major issues

Lines 286, 314, 316, 318, 324 and more?

Please double-check the values, since it appears that you have used a point (full stop, period) instead of a comma. This is potentially very confusing and reduces reader confidence in your paper.

e.g. 5.141 - - > 5,141

Minor issues

Line 26

Please support your claim of “half” with a suitable reference. This article whose link is below puts the proportion at around a third in the US.

https://www.cell.com/cell/fulltext/S0092-8674(17)31384-3

Lines 23, 29, 42 and many more

Use consistent format for paragraphs, i.e. indent new paragraphs.

Line 45

Provide full form for abbreviated forms.

Lines 66-69

I assume that Steps 4 and 5 are future work (rather than salami slicing your research output). If so, perhaps it is worth stating that here.

Lines 283-4

Please explain why you selected those two pathways.

Table 9 (b)

This table needs to be formatted correctly.

Minor language issues

Title [case]

I think that this journal requires titles to be written in title case.

Line 4

ABLISS project whose the objective is literature-based discovery, more precisely whose the objective is - - > ABLISS, literature-based discovery project, whose objective is

Line 6 [clarity]

from these facts  - - > from the extracted facts

Lines 7-8 [capitalization and one typo]

The definition of our model of predicates; The NLP programm using Finite State Methods (Unitex graph cascades); And - - > the definition of our model of predicates; the NLP program using Finite State Methods (Unitex graph cascades); and

Line 9

the whole interesting papers - - > I suspect that interesting is not the most appropriate adjective here.

Line 51 [confusing]

close work  - - > Perhaps, you mean related work?

Line 51-2 [clarity]

In this last paper- - > In the latter,

Line 60 [ambiguity]

related  - - > described

Line 69

present the continuation  - - > describe the planned future work

Line 85

allow one for making   - - > allow the making of hypotheses?  

Line 93

shown Figure 1  - - > shown in Figure 1

Line 94, 102, etc. [formality]

The contracted form is rather informal, so I suggest avoid using contractions.

didn’t   - - > did not

doesn’t - -> does not

Line 153 [formality]

A bit  - - > rather

Line 183 [confusing]

resp.  - - > Provide the full form.

Table 1

Consider transforming the sentence that runs on after the title into a footnote for the table.

Line 216

when using  - - > when used OR when being used

Line 272

on a list  - - > based on a list?

Table 3

Images do not quite fit into the cells in the table. Reformat.

Line 299

After these operations, it remained 548 papers to analyze.  - - > This reduced the number of papers to be analyzed to 548.

Line 311

the cascade allowing to construct the biological predicates  - - > the cascade to construct the biological predicates OR the cascade allowing the construction of the biological predicates

Line 313

this allows - - > which allows us

Line 319

sixteen next graphs - - > next sixteen graphs

Line 345

Omit the errant bracket.

Line 352

Why is Figure 10 described before Figure 9? Renumber the figures so that they occur sequentially in the text.

Line 506

negative false - - > false negatives

Line 519

bad recall - - > poor recall score

Line 547

allows one - - > allows

Line 567

extraction, are - - > extraction are

Line 609 and 4 [consistency]

Select one form for the project name: ABLISS or Abliss.

Reviewer 3 Report

This work describes the first phases of the project ABLISS, whose objective is to extract network information from scientific literature about G-protein coupled receptors (GPRCs) and use automated deductive reasoning to build networks from the facts extracted. More precisely, this paper focuses on three main tasks: a) formalization of biological facts as predicates and rules, b) extraction of predicates from scientific papers with Unitex graph cascades, and c) design of a predicates and rules database for the next phase of deductive reasoning.

(1) There is not a section in the paper devoted to related works. There is only a single paragraph in the introduction, referencing a handful of papers, mostly from the 2000s. Since the kind of knowledge extraction described in this paper does not seem very novel to me, I would expect more relevant (and recent) previous works to exist, and I like those publications to be addressed in a proper "Related work" section.

(2) The authors describe in section 4 the design of the relational database used to store the predicates and rules that will be used for inference. Though the database does not seem very complicated, an Entity Relationship Diagram, which many modern database management systems generate more or less automatically, would give the reader a faster understanding of the database design.

(3) The authors test the precision and recall of their processing chain using two papers. It is not explicitly said how the gold standard of predicates was extracted from those papers. I assume it was done manually. If it was, how many experts were involved? Was there total agreement or some amount of disagreement between them? That information is relevant to assess the quality of the tests conducted, and should be included. In addition, the paper does not list the number of predicates to be found (manually or otherwise) in each paper. If the number is too small, the results obtained may not be very representative. I would like to see this information in the text, and maybe some additional testing if it turns out that there were only a few predicates to extract.

(4) While the low recall in the extraction of relation and action predicates should be ameliorated by a more complete list of predicates, the authors do not explain how the low precision in the extraction of ontological predicates can be addressed, or if they plan to do so.

(5) In most of the figures, the text is small almost to the point of being illegible. Font size, graph size or both should be increased.

(6) While the paper is, in general, intelligible and well written, there are some mistakes in the English used. Without being necessarily exhaustive:

- Abstract: "We present here the ABLISS project whose the objective is literature-based discovery, more precisely whose the objective is to be able to extract network information from literature using Natural Language Processing (NLP), then build networks from these facts using automated deductive reasoning.". I am not a native English speaker, but "whose the objective" do not seem correct to me. In addition, two occurrences of "whose the objective" that close to one another is a little redundant.
- Page 3, paragraph 3: I do not understand what "close work" is. Do the authors mean "closely related work"?
- Page 3, second bullet point: "Formalism should be chemically precise...". Should not the word "formalism" be preceded by "The"?
- Page 4, first bullet point: "The formalism has to allow one for making hypotheses." One what? Do the authors mean that the formalism has to allow for hypothesis making?
- Page 4, first paragraph of subsubsection 2.1.1: "the different components of the system, shown Figure 1." I think there is a "in" missing before "Figure 1".
- Page 16, first paragraph of subsection 3.4: "..two papers on which we had not work...". I think it should be "worked" instead of "work".
- Page 17, fist paragraph: "we have not negative false" should be "we have not found false negatives" or, even better, "the chain has not produced false negatives".
- Page 17, instance 2: I think "the implemented predicate reationModulationMediation are not found because the graph not described this context" should be "the implemented predicate reationModulationMediation IS not found because the graph DOES not DESCRIBE this context".

A thorough proof reading would enhance the final manuscript.

Round 2

Reviewer 1 Report

The authors improved the quality of the paper and it can be acceptable for publication.

Reviewer 2 Report

Comments and suggestions for authors

Thank you for the detailed explanation of the changes made based on reviewer suggestions and for providing cogent explanations for changes that you did not or could not implement

This version of your paper is much improved.

You have addressed the concerns that I raised in my initial review.

Some of the figures are still difficult to read, but given that software does not allow you to increase the font size, this is unresolvable. It is still possible to make out what is written so that is fine.

I have no other recommendations although I did notice another typo that you should perhaps change before submitting the final camera-ready version.

Typo

Line 393

one of our graphs as been designed - - > one of our graphs has been designed